# Additional Benefit of Intraoperative Electroacupuncture in Improving Tolerance of Deep Brain Stimulation Surgical Procedure in Parkinsonian Patients

**DOI:** 10.3390/jcm11102680

**Published:** 2022-05-10

**Authors:** Sylvie Raoul, Régine Brissot, Jean-Pascal Lefaucheur, Jean-Michel Nguyen, Tiphaine Rouaud, Yunsan Meas, Alain Huchet, Ndrianaina Razafimahefa, Philippe Damier, Julien Nizard, Jean-Paul Nguyen

**Affiliations:** 1Service de Neurochirurgie, Hôpital Laennec, CHU, 44093 Nantes, France; rbrissot58@aol.com (R.B.); ndrianaina@yahoo.fr (N.R.); 2EA4391, Excitabilité Nerveuse et Thérapeutique, Université Paris Est Créteil, 94000 Créteil, France; jplefaucheur@aphp.fr (J.-P.L.); julien.nizard@chu-nantes.fr (J.N.); 3Unité de Neurophysiologie Clinique, Hôpital Henri Mondor, AP-HP, 94000 Créteil, France; 4Service de Biostatistiques et d’épidémiologie, Hôpital Saint Jacques, CHU, 44093 Nantes, France; jeanmichel.nguyen@chu-nantes.fr; 5Service de Neurologie, Hôpital Laennec, CHU, 44093 Nantes, France; tiphaine.rouaud@chu-nantes.fr (T.R.); philippe.damier@chu-nantes.fr (P.D.); 6Service Douleur, Soins palliatifs et de Support et UIC22, Hôpital Laennec, CHU, 44093 Nantes, France; docteur@meas.fr (Y.M.); jeanpaul.nguye@gmail.com (J.-P.N.); 7Faculté de médecine, CHU, 44093 Nantes, France; alain.huchet@free.fr; 8Centre D’évaluation et de Traitement de la Douleur, Clinique Brétéché, Groupe Elsan, 44000 Nantes, France

**Keywords:** acupuncture, comfort, deep brain stimulation, delirium, electroacupuncture, sickness behavior

## Abstract

Background: Deep brain stimulation (DBS) is an effective technique to treat patients with advanced Parkinson’s disease. The surgical procedure of DBS implantation is generally performed under local anesthesia due to the need for intraoperative clinical testing. However, this procedure is long (5–7 h on average) and, therefore, the objective that the patient remains co-operative and tolerates the intervention well is a real challenge. Objective: To evaluate the additional benefit of electroacupuncture (EA) performed intraoperatively to improve the comfort of parkinsonian patients during surgical DBS implantation. Methods: This single-center randomized study compared two groups of patients. In the first group, DBS implantation was performed under local anesthesia alone, while the second group received EA in addition. The patients were evaluated preoperatively, during the different stages of the surgery, and 2 days after surgery, using the 9-item Edmonton Symptom Assessment System (ESAS), including a total sum score and physical and emotional subscores. Results: The data of nine patients were analyzed in each group. Although pain and tiredness increased in both groups after placement of the stereotactic frame, the ESAS item “lack of appetite”, as well as the ESAS total score and physical subscore increased after completion of the first burr hole until the end of the surgical procedure in the control group only. ESAS total score and physical subscore were significantly higher at the end of the intervention in the control group compared to the EA group. After the surgical intervention (D2), anxiety and ESAS emotional subscore were improved in both groups, but the feeling of wellbeing improved in the EA group only. Finally, one patient developed delirium during the intervention and none in the EA group. Discussion: This study shows that intraoperative electroacupuncture significantly improves the tolerance of DBS surgery in parkinsonian patients. This easy-to-perform procedure could be fruitfully added in clinical practice.

## 1. Introduction

Deep brain stimulation (DBS) was developed by Alim Benabid et al. in 1987 [1] for the treatment of patients with Parkinson’s disease (PD) and its efficacy has now been widely confirmed [2,3,4,5]. The technique [1,2] involves placing electrodes in the dorsolateral part of the subthalamic nucleus (STN), usually on both sides. The STN target is primarily identified by a neuroimaging approach, e.g., based on T2-weighted MR images fused with stereotactic CT scan. In addition to imaging-guided target location, the surgical procedure of DBS electrode implantation may also benefit from electrophysiological recordings of neuronal activities and clinical testing. Intraoperative recordings using microelectrodes are able to improve the accuracy of reaching the target and then the clinical outcome of the procedure [6]. Such recordings can be performed under general anesthesia [7], though even this induces some disturbances [8]. On the other hand, intraoperative clinical testing requires local anesthesia and a conscious patient. Actually, when the DBS electrode is correctly positioned, the improvement of the main parkinsonian symptoms, such as tremor, hypertonia, and akinesia, by high-frequency stimulation can be immediately verified intraoperatively. Adverse effects can also be detected, e.g., paresthesia when the tip of the electrode is too medial and posterior, or muscle contraction when too lateral. Owing to the importance of clinical testing, most authors recommend using local anesthesia for DBS electrode implantation [1,9,10]. However, this surgical procedure applies to parkinsonian patients who must be “off-medication” and immobilized on the operating table for a very long time, often between 5 and 7 h [11]. Therefore, some patients had clouding of consciousness or became confused, unable to follow neurologist instructions for clinical testing. In the literature, intraoperative alteration of consciousness occurs in 0.9 to 14.6% of cases [11,12,13,14,15,16,17]. In addition, some patients may have neurovegetative reactions, with nausea, vomiting, or high blood pressure, or show anxiety disorders, with agitation or panic attack, which leads to interrupting the surgery. This situation was shown to occur in 0.5 to 2.1% of cases [11,12,15,16] and was described in the literature as intraoperative “sickness behaviour” (SB) or delirium [14,17,18]. Finally, such an adverse event has been reported to persist during the postoperative period in 2.8 to 33.3% of cases [12,19,20,21,22,23], increasing the risk of severe complications (pneumopathy, pulmonary embolism, etc.) [14,18,23,24].

Studies evaluating strategies to reduce the occurrence of such a deleterious intraoperative situation have been rarely performed. We thought that acupuncture could be one of those strategies. Acupuncture analgesia causes conscious sedation and helps limit the development of neurovegetative symptoms. Acupuncture analgesia has been extensively used in different surgical fields [25], such as abdominal [26], gynecological [27], and thoracic surgery [28], as well as in neurosurgery [29,30,31]. In most studies, acupuncture is combined with conventional anesthesia before, during, and after surgery [32]. These studies showed that anesthesia may be lighter with acupuncture, with less intraoperative consumption of anesthetic drugs [33], and that comfort was improved in the postoperative period [26], with less opioid consumption.

Acupuncture analgesia has also been used during resection of brain lesions (e.g., tumors) located in eloquent areas. As with DBS surgery, the patient must be conscious during this type of surgery in order to perform clinical testing essential to prevent damage to language or motor functions. In this context, acupuncture analgesia has been used alone [34], in association with local anesthesia [35], or with light general anesthesia [36]. Because all these articles are in Chinese, they are difficult to read in detail. They do not seem to mention any occurrence of SB or delirium in either the acupuncture or control groups, but the patients were young, with a low risk of developing intraoperative behavior disorders.

To our knowledge, this is the first study to assess the value of acupuncture analgesia, consisting of an electroacupuncture (EA) procedure, in improving the comfort (including the sensation of wellbeing) of parkinsonian patients undergoing DBS surgery, as well as to prevent intra- or postoperative occurrence of SB or delirium.

## 2. Methods

### 2.1. Study Design

This monocentric randomized study had a parallel-group design, comparing the use of EA associated with local anesthesia to local anesthesia alone. Ten patients per group were included, according to sample size calculation based on a decrease in the ESAS score of more than 50% in 80% in the experimental group against 20% in the control group, with an alpha risk of 5% and a power of 80%. The study was performed at the Nantes University Hospital (Département de Neurochirurgie et Centre d’Évaluation et de Traitement de la Douleur), approved by the local Ethics Committee (CCP West V No. 14/23-939 on 10/4/2014), and registered on ClinicalTrials.gov (NCT02236260).

Inclusion criteria were:Parkinsonian patients of both sexes with age ≥18 years.Scheduled DBS intervention targeting the STN.Signed informed consent.

Exclusion criteria were:
Patients with age ≥ 70 years.History of intolerance to acupuncture.Unstable psychiatric disorder.Legal protection regimen.

### 2.2. Assessment

The primary endpoint was patient’s comfort, as assessed by the Edmonton Symptom Assessment System (ESAS) scale [37]. This scale included 9 items (pain, tiredness, drowsiness, nausea, lack of appetite, shortness of breath, depression, anxiety, and wellbeing) and a final open question about other possible symptoms (in particular, regarding consciousness or cognitive impairment). For each item, the score is expressed on a visual analogue scale (VAS) from 0 to 10, with 0 corresponding to no impairment and 10 to maximal impairment [38,39]. Beyond the total sum score of the 9 items, physical and emotional ESAS subscores are defined, including 6 items (pain, tiredness, drowsiness, nausea, lack of appetite, and shortness of breath), and 2 items (depression and anxiety), respectively [40]. Since its first development in 1991 for the assessment of palliative care patients, ESAS has been validated in many languages and is now commonly used by many groups worldwide for multiple symptom screening and longitudinal monitoring in many disciplines. This scale is a clinically pragmatic, quick, and easy-to-administer assessment tool to simultaneously document the impact of an intervention on different symptom clusters [40].

The secondary outcome measures were:Intraoperative anesthesia monitoring parameters: heart rate, oxygen saturation, and systolic and diastolic blood pressure.Total time duration of the surgery.

Age, gender, unified Parkinson’s disease rating scale (UPDRS) part III motor subscale [41] and Hamilton Anxiety Rating Scale (HAMA) were recorded at inclusion to analyze whether the intervention and control groups differed in terms of baseline demographic characteristics and clinical profile. Finally, all intraoperative and postoperative adverse events were collected.

### 2.3. Acupuncture Procedure

Acupuncture was performed bilaterally on the following acupoints (Figure 1): Hegu LI 4 and Neiguan P 6 using surface acupuncture electrodes, and Taichong LV 3 and San yinjiao SP 6 using classic Dong Bang needles (diameter: 0.25 mm; length: 62 mm).

Electrostimulation was administered with an Algistim Duo^®^ device (Sédatelec, Irigny, France), with 4 independent channels, 1 for each of the patient’s limbs. This makes it possible to connect 8 needles or self-adhesive electrodes. Stimulation parameters consisted of asymmetric rectangular pulses of 0.5 ms duration with a frequency varying from 1 to 99 Hz and including an automatic sweep within this range of frequencies as a discontinuous wave. Stimulation intensity was adapted to the patient’s perception, at the limit of the pain threshold, and, if possible, provoking a motor reaction. These are the generally accepted parameters for optimal analgesia [42,43].

The choice of the acupuncture points was based on the fact that these points belong to a common set of analgesic acupuncture points, notably used in surgery, and frequently cited in publications in this context [25,26,27,42,43]. In addition to analgesia, their other potential actions are effects on the autonomic nervous system, in particular nausea and vomiting (Neiguan P 6), headaches (Hegu LI 4, Taichong LV 3), or sleep (Sanyinjiao SP 6).

Patients included in the EA group were given a detailed description of the intraoperative EA procedure and a first session was performed the day before the surgery. 

The same EA protocol was applied in the operating room as soon as the patient was resting on the operative table. EA was maintained throughout the procedure, except: i) when transferring the patient to the radiology room for the stereotactic CT scan, ii) during the electrophysiological recording of neuronal activities to avoid electromagnetic interference, and iii) during the clinical testing period because of the risk of displacing the acupuncture needles during active and passive movements of the upper limbs.

In both groups, local anesthesia (lidocaine 1%) was used to avoid pain during the fixation of the stereotactic frame on the skull and incisions of the scalp for completion of the burr holes. Local anesthesia was occasionally associated with the inhalation of an equimolar mixture of oxygen and nitrogen oxide (MEOPA^®^) depending on the patient’s reaction.

### 2.4. Statistical Methods

Two-way repeated-measures ANOVA was performed to analyze scores of each ESAS item, as well as total ESAS score and physical and emotional ESAS subscores recorded in the two groups of patients (control and EA groups) at the following timepoints:Baseline/inclusion (incl): the day before the surgery.Positioning/installation (inst): a few minutes after the patient rested quietly on the operating table.After placement of the stereotactic frame (post SF).Immediately after completion of the first burr hole (post BH1).Immediately after completion of the second burr hole (post BH2).One hour after completion of the second burr hole (1H post BH2).After ablation of the stereotactic frame at the end of the procedure (End).2 days after surgery (D2).

The missing data were handled by using the last observation carried forward method. The significance threshold was set at 5%. In case of significance of time interaction, Dunnett’s multiple comparisons post hoc test was used, comparing data recorded at all the timepoints in each group (control or EA) with respect to the initial timepoint (incl) taken as a reference. In case of significance of group interaction, Sidak’s multiple comparisons post hoc test was used to compare the data recorded in the two groups (EA and control) at all the timepoints.

As secondary outcome measures, intraoperative anesthesia monitoring parameters (heart rate, oxygen saturation, and systolic and diastolic blood pressure) were analyzed with the same repeated-measures ANOVA, but only considering intraoperative time points (from inst to end).

A comparison between the two groups (control and EA) at baseline was performed using the Mann–Whitney test for age, UPDRS score, total ESAS score, and physical and emotional ESAS subscores, and the Fisher’s exact test for gender. A comparison between the two groups regarding the total time duration of the surgery was also performed using the Mann–Whitney test. Prism statistical software (GraphPad Software, San Diego, CA) was used in all cases.

## 3. Results

The two groups of patients were similar for age, gender, and PD motor severity at baseline (Table 1). There was also no significant difference in the level of patient’s comfort (ESAS score) between the two groups before the surgery. In contrast, anxiety (HAMA score) was significantly lower in the control group than in the EA group the day before surgery (Table 1).

Two patients were excluded from the study. One patient withdrew his consent because he did not want acupuncture and the other, who was randomized in the control group, presented with invalidating sciatic neuralgia, which required acupuncture at the time of surgery (Figure 2).

Two-way repeated-measures ANOVA showed a significant time interaction for the following ESAS items: pain (F (DFn = 7, DFd = 56) = 9.26, η^2^ = 0.28, *p* < 0.0001), tiredness (F (7, 56) = 17.56, η^2^ = 0.35, *p* < 0.0001), drowsiness (F (7, 56) = 2.85, η^2^ = 0.10, *p* = 0.013), lack of appetite (F (7, 56) = 6.26, η^2^ = 0.14, *p* < 0.0001), shortness of breath (F (7, 56) = 2.33, η^2^ = 0.02, *p* = 0.037), anxiety (F (7, 56) = 11.04, η^2^ = 0.26, *p* < 0.0001), and wellbeing (F (7, 56) = 8.85, η^2^ = 0.23, *p* < 0.0001), as well as for the total ESAS score (F (7, 56) = 18.21, η^2^ = 0.30, *p* < 0.0001), physical subscore (F (7, 56) = 18.69, η^2^ = 0.32, *p* < 0.0001), and emotional subscore (F (7, 56) = 9.58, η^2^ = 0.24, *p* < 0.0001). Thus, the effect size was large (η^2^ ≥ 0.14) in all cases, but medium or small for the items “drowsiness” and “shortness of breath”, respectively.

Post hoc tests showed a significant score increase compared to incl scores for:-“pain”, from post SF to end in both groups (mean (95% confidence interval) difference = 1.24 (−0.80, 3.29) to 4.43 (2.39, 6.48));-“tiredness”, from post BH2 or 1H post BH2 to end in both groups (2.93 (−0.43, 5.43) to 5.69 (3.19, 8.19));-“lack of appetite”, from post SF to end in the control group only (4.19 (1.47, 6.91) to 5.04 (2.32, 7.77));-ESAS total score, from post BH1 to end in the control group only (10.79 (1.29, 20.29) to 18.07 (8.57, 27.57));-ESAS physical subscore, from post BH1 to end in the control group only (9.51 (2.59, 16.43) to 17.40 (10.48, 24.32)) (Figure 3).

Post hoc tests showed a significant score decrease compared to incl scores for:-“anxiety” at D2 in both groups (−2.51 (−0.29, −4.73) to −2.70 (−0.48, −4.92));-ESAS emotional subscore at D2 in both groups (−2.66 (−0.06, −5.25) to −2.70 (−0.11, −5.29));-“wellbeing” at D2 in the EA group only (−2.43 (−0.27, −4.60)) (Figure 3).

Significant group interaction (F (1, 8) = 10.06, η^2^ = 0.14, *p* = 0.013) and time × group interaction (F (7, 56) = 5.09, η^2^ = 0.12, *p* = 0.0002) were observed for the item “lack of appetite” only (at the limits of a large effect size). Post hoc tests showed a significant score increase in the control versus EA group from post BH1 to end for this item (mean (95% confidence interval) difference = 3.87 (1.01, 6.73) to 4.79 (1.93, 7.65)), and also at the end of the intervention for ESAS total score (10.24 (0.26, 20.23)) and physical subscore (9.28 (2.00, 16.55)) (Figure 3).

Regarding the intraoperative anesthesia monitoring parameters (heart rate, oxygen saturation, and systolic and diastolic blood pressure) no significant change during the surgical intervention was observed within or between groups using repeated-measures ANOVA (*p* > 0.05 for time, group, or time x group interactions in all cases). Finally, the total time duration of the surgery did not differ between the two groups (mean: 6:17 in the control group vs. 6:53 in the EA group, *p* = 0.27).

Regarding adverse events, one patient in the control group developed delirium associated with a panic attack immediately after the placement of the first electrode and before the completion of the second burr hole, so that the intervention was stopped. No adverse event was observed in the EA group.

## 4. Discussion

This study shows that EA can reduce the discomfort of the surgical procedure of DBS implantation performed under local anesthesia in PD patients. To summarize, pain and tiredness increased in both groups after placement of the stereotactic frame, but the score of the ESAS item “lack of appetite”, as well as both ESAS total score and physical subscore, increased only in the control group from the completion of the first burr hole until the end of the intervention. At that time point, these scores were significantly higher in the control group than in the EA group. All these results show greater discomfort during and at the end of the intervention in the control group compared to the EA group.

In the control group, the ESAS item “lack of appetite” was the main sign of discomfort, followed by tiredness and pain, integrating well into the clinical picture of SB [44]. ESAS physical subscore increased markedly in the control group from the completion of the first burr hole, indicating physical discomfort, while the ESAS emotional subscore remained relatively stable. This suggests that, during the intervention, patients need more help to reduce physical discomfort than psychological support and that this goal can be achieved through the use of EA.

After the surgical intervention (D2), anxiety and ESAS emotional subscores were reduced in both groups, while the feeling of wellbeing only improved in the EA group (a lower score corresponding to a better status). In the control group, one patient developed delirium after placement of the first electrode, leading to discontinuation of the procedure. This case could have been caused by direct damage to the limbic STN or be due to any kind of concurrent surgical factor. Therefore, no conclusion can obviously be drawn from this single case. In a second time, this patient underwent DBS electrode implantation using EA without any adverse event.

In clinical practice, DBS is indicated in PD patients who are under the age of 70 with a Mattis score < 130/144 [45]. Patients who are near these limits could be a population at risk of delirium, although the results in the literature are sometimes contradictory [5,17,23]. Our results suggest that EA significantly improves the patient’s comfort, especially in the last part of the procedure, between 5 and 6 h after patient’s installation, when tiredness and physical discomfort are at their peak.

The development of SB may be linked to an increase in the production of proinflammatory cytokines [46]. The anti-inflammatory action of acupuncture has been shown in several studies [47,48], including the reduction in interleukin-6 (IL-6) and interleukin-1β (IL-1β) serum levels in a surgical context [49,50]. Acupuncture may also reduce the development of emotional and attention disorders by reducing stress-induced inflammatory reactions [51]. Therefore, it would have been interesting to take blood samples of proinflammatory cytokines (IL-1β and IL-6 in particular), before, at different points during, and after the procedure to identify the possible anti-inflammatory effects of EA.

One limitation of EA in the context of DBS implantation surgery is to maintain the procedure during clinical testing of the PD patients because it requires active and passive movements of the distal upper limbs, at the level of which there were two acupuncture points used in this protocol. The value of other acupuncture points located in the leg should be assessed, such as the Zu San Li acupoint (St 36) [25], which are known also to produce analgesic and anti-inflammatory effect [47].

However, the main limitation of this study is the absence of blinding or a sham procedure. The absence of sham intervention is a common ethical limitation of surgical interventions. On the other hand, it is obvious that it is not possible to make patients “blind” to EA treatment. The feasibility of blinding and sham intervention in randomized controlled clinical trials of acupuncture has been widely discussed [26], considering the technical and physiological aspects of acupuncture. Thus, patients in the EA group may have felt better simply because they “knew” they were being treated with EA, not because of the EA intervention per se. However, in our study, patients in the EA group were significantly more anxious before the operation than patients in the control group. It is possible that the explanations given to the patients about the EA procedure initially increased their anxiety. This would reinforce the hypothesis of a real efficacy of EA rather than a “care” effect in improving patient comfort during surgery. In any case, a head-to-head comparison of two EA procedures, relevant and irrelevant, could, for example, be a good approach in a future study to confirm the present results.

It is important to also consider the potential subjectivity of a questionnaire for the clinical evaluation of patient comfort. In the field of DBS [15,17], but also in craniotomies performed in awake patients [42], intraoperative assessment is often limited to scoring pain intensity on a one-dimensional scale or reporting adverse events. In humans, there are quantified scales to assess SB [44,52], but these are usually adapted to the context of everyday life, with questions regarding libido, feelings of pleasure, loss of weight, sleep quality, and social isolation, which are not adapted to the intraoperative context [53,54]. This was one of the reasons for using the ESAS and its sub-scores, suitable for the evaluation of an intervention and whose use has been validated, including outside the context of palliative care, in which ESAS was originally developed [40].

Finally, other major limitations of the study are the small sample size and the single-center design. Considering the other limitations mentioned above, such as the absence of blinding or sham procedure, this study should, therefore, be considered as a preliminary study whose results must be confirmed in future studies including a more stringent control procedure and larger sample size.

## 5. Conclusions

This study shows that EA improves intra- and postoperative comfort of PD patients who undergo surgical implantation of DBS electrode. This improvement was mainly found during the second part of the surgery, which is generally quite difficult for the patient, as well as during the early postoperative period (first 48 h). Our results suggest that acupuncture should be more systematically proposed to patients presenting with risk factors of delirium due to awake neurosurgery, but remain to be confirmed in a multicenter trial with a larger group of patients.

## Figures and Tables

**Figure 1 jcm-11-02680-f001:**
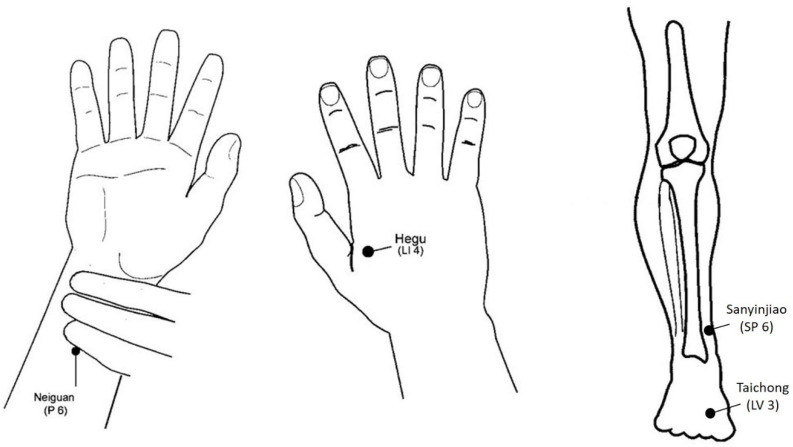
Location of the acupuncture points.

**Figure 2 jcm-11-02680-f002:**
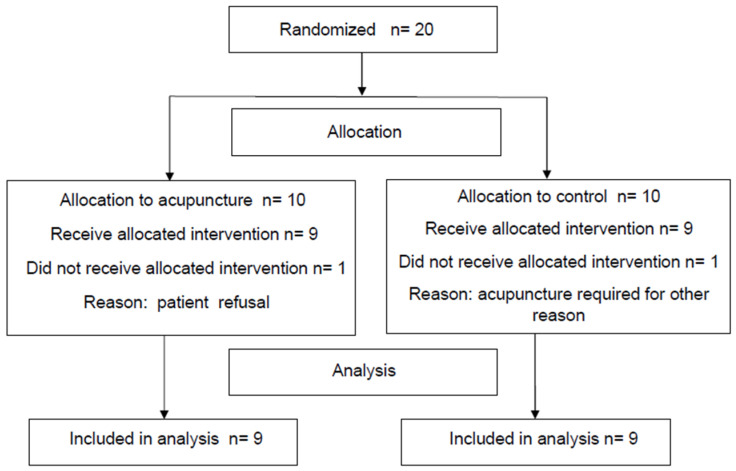
Flow chart.

**Figure 3 jcm-11-02680-f003:**
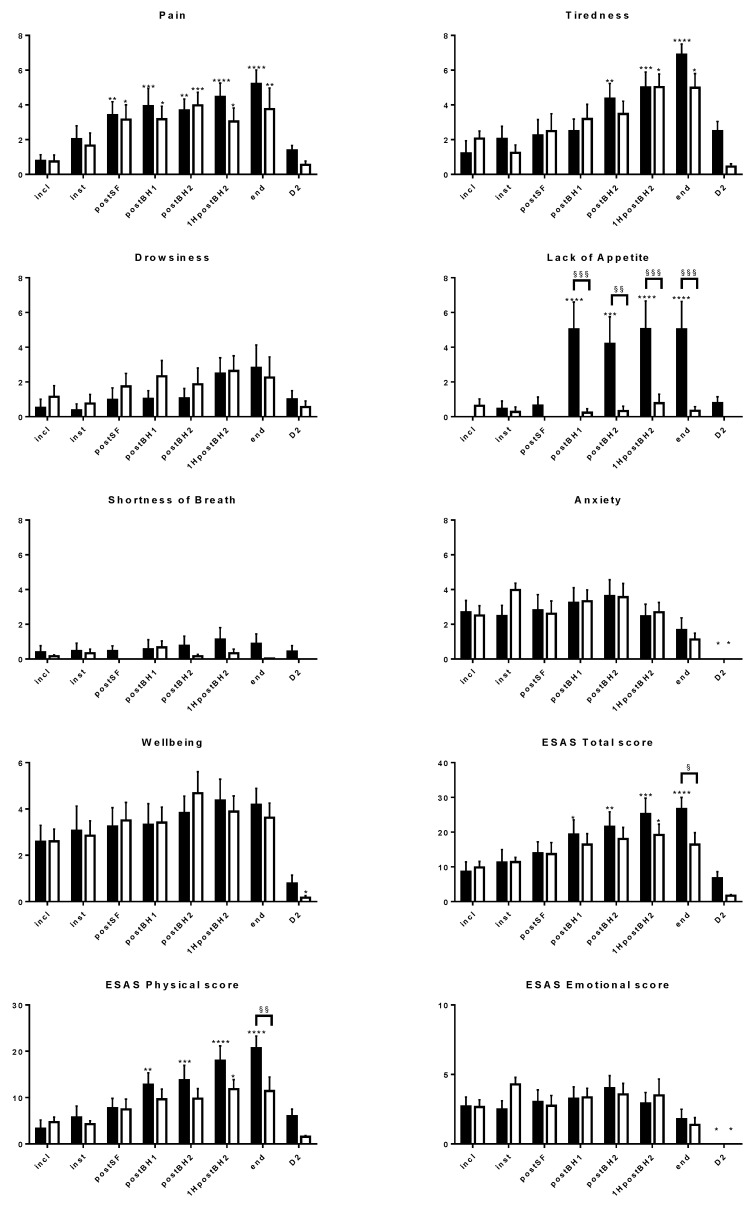
Time course of the ESAS variables. Only variables for which a significant time or group interaction was found (repeated-measures ANOVA, *p* < 0.05) are reported. In black: control group. In white: electroacupucture group. Post hoc tests for score change compared to incl score: *, *p* < 0.05; **, *p* < 0.01; ***, *p* < 0.001; ****, *p* < 0.0001. Post hoc tests for score difference between the control and EA groups at a given timepoint: §, *p* < 0.05; §§, *p* < 0.01; §§§, *p* < 0.001. incl, inst, postSF, postBH1, postBH2, 1HpostBH2, end, D2: various timepoints before, during, or after the surgery (see text).

**Table 1 jcm-11-02680-t001:** Description of population of the study.

	Control Group (*n* = 10)	EA Group (*n* = 10)	*p* Values
Age (years) mean (sd)	56 (7.1)	57 (5.7)	0.40
Gender (women. men)	5w. 5m	2w. 8m	0.33
UPDRS-III score mean (sd)	22 (11.2)	22 (13.3)	0.56
HAMA score (sd)	7.6 (1.9)	12.4 (2.4)	0.009
ESAS total score mean (sd)	8.6 (8.6)	9.8 (5.2)	0.38
ESAS physical subscore mean (sd)	3.3 (5.6)	4.7 (3.2)	0.10
ESAS emotional subscore mean (sd)	2.7 (2.0)	2.6 (1.6)	0.79

## Data Availability

The data that support the findings of this study are available on request from the corresponding author.

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
