# Peer review of "Additional Benefit of Intraoperative Electroacupuncture in Improving Tolerance of Deep Brain Stimulation Surgical Procedure in Parkinsonian Patients"

_jcm, 2022, doi:10.3390/jcm11102680_

Round 1

Reviewer 1 Report

This study presents results of additional electroacupuncture in patients with Parkinson’s disease undergoing stereotactic deep-brain stimulation surgery. The study aimed to show differences in tolerance of surgical procedure between the intervention and the treatment-as-usual group. The article is well written and of high clinical importance. The study advances the field of DBS surgery as the integration of acupuncture analgesia in DBS surgery seems to be a good choice to ameliorate intraoperative sickness behaviour. Though the results are interesting, they can only be regarded as preliminary and the manuscript could benefit from some improvements.

Minor comments:

Page 2, line 52: Please add a full-stop after the “(8)” bracket.

Page 2, line 68: Please add a full-stop after the “(14, 18, 17)” bracket.

Page 2, line 70: double-check formatting in the “(19, 20…23)” bracket: comma is missing, spaces are doubled, bracket at the end is missing.

Page 2, line 71: Please add a full-stop after the “(14, 18…)” bracket. Double-check formatting of the numbers in bracket (comma, double-spaces).

Page 2, line 74: please check double-spaces after “analgesia”

Page 2, line 86: there is a reference missing in empty brackets

Please double-check spaces throughout the manuscript (I did not indicate all instances (see brackets in the discussion section…).

Methods

Study design:

Page 3, line 110: I recommend formatting this line and highlight the exclusion criteria just as the inclusion criteria with a new subheading.

Assessment:

Authors used the Edmonton Symptom Assessment System (ESAS) to evaluate intraoperative sickness behavior. The ESAS is a common instrument to assess symptom burden in cancer patient receiving palliative care. The use of the ESAS during DBS surgery requires some rationale to follow. Please, give some reason why this instrument was chosen for this specific study.

Page 3, line 127: Authors mention age and gender as secondary outcome measures. Please clarify, why age and gender were used as outcome measures? Usually, age and gender are used as covariates. Maybe, I am just wondering about the expression “outcome measure”. For me, it seems, authors just analysed whether intervention and control groups differed in terms of age, gender and baseline symptomatology. I do not consider these factors as outcome measures, but rather (sociodemographic) characteristics.

Page 3, line 136: please follow one formatting style for the names of the acupuncture points (either with or without space between channel and number (e.g. “LI 4” and “P6”) see also Figure 1 as reference).

Page 3, line 136: Please, separate “Sanyinjiao” after one Chinese word “San yin jiao” (e.g. “San- yinjiao” instead of “Sa-nyinjiao”).

Page 3, line 137: Please, give the correct diameter of the acupuncture needles (probably “0.25mm” and not “25mm”).

I recommend giving some rationale for the choice of the acupuncture points. Is the point selection a common set of acupuncture points for acupuncture analgesia or was it chosen for any particular reason? This could be interesting to the reader.

Results:

Page 5, line 194: authors mention the HAMA score. The HAMA (probably Hamilton Anxiety scale) is not mentioned in the methods section. Please, clarify the use of the HAMA in the methods section.

Page 5, line 219: I wondered whether the word “differences” is missing in this sentence? Do the authors mean to say “Significant group and time x group interaction differences were only observed for the item ‘lack of appetite’”?

Discussion:

Page 8, line 248: I recommend to give some interpretation of the ESAS score. What does increase of the total or a subscore mean? E.g. “…there was a significant score increase in the control versus EA group at the end of the intervention for ESAS total score and physical subscore indicating higher symptom in the control group.

Page 8, line 249: Authors state that wellbeing increased only in the EA group. Figure 3 indicates lower wellbeing scores in the EA group compared to the control group at D2. Does a lower wellbeing score indicate higher wellbeing? This is why I recommend giving some kind of interpretation of the scores for those readers who are not familiar with the ESAS score (as I am).

Page 8, line 250: Please, double-check the sentence starting with “Finally, one patient developed…”. This sentence seems incomplete.

Page 8, line 251: Authors draw the conclusion that EA increases the probability of completing DBS procedure as one patient developed delirium during DBS surgery in the control group and the intervention was stopped. I do not follow this statement as the number of n=1 is too small to draw any firm conclusions. Please, reword this sentence to a more cautious conclusion.

The authors should list some more limitations of the study at hand such as small sample size, single study center, lack of blinding, lack of a sham intervention, lack of a firm primary hypothesis being tested…

Author Response

English language and style

( ) Extensive editing of English language and style required
( ) Moderate English changes required
(x) English language and style are fine/minor spell check required
( ) I don't feel qualified to judge about the English language and style

Yes

Can be improved

Must be improved

Not applicable

Does the introduction provide sufficient background and include all relevant references?

(x)

( )

( )

( )

Is the research design appropriate?

( )

(x)

( )

( )

Are the methods adequately described?

( )

( )

(x)

( )

Are the results clearly presented?

(x)

( )

( )

( )

Are the conclusions supported by the results?

( )

( )

(x)

( )

Comments and Suggestions for Authors

This study presents results of additional electroacupuncture in patients with Parkinson’s disease undergoing stereotactic deep-brain stimulation surgery. The study aimed to show differences in tolerance of surgical procedure between the intervention and the treatment-as-usual group. The article is well written and of high clinical importance. The study advances the field of DBS surgery as the integration of acupuncture analgesia in DBS surgery seems to be a good choice to ameliorate intraoperative sickness behaviour. Though the results are interesting, they can only be regarded as preliminary and the manuscript could benefit from some improvements.

Minor comments:

Page 2, line 52: Please add a full-stop after the “(8)” bracket. It was corrected

Page 2, line 68: Please add a full-stop after the “(14, 18, 17)” bracket. It was corrected

Page 2, line 70: double-check formatting in the “(19, 20…23)” bracket: comma is missing, spaces are doubled, bracket at the end is missing. It was corrected

Page 2, line 71: Please add a full-stop after the “(14, 18…)” bracket. Double-check formatting of the numbers in bracket (comma, double-spaces). It was corrected

Page 2, line 74: please check double-spaces after “analgesia” It was corrected

Page 2, line 86: there is a reference missing in empty brackets . It was corrected

Please double-check spaces throughout the manuscript (I did not indicate all instances (see brackets in the discussion section…).I try to do it carefully and correct everything I see

Methods

Study design:

Page 3, line 110: I recommend formatting this line and highlight the exclusion criteria just as the inclusion criteria with a new subheading. You’re right and I have maked the change.

Assessment:

Authors used the Edmonton Symptom Assessment System (ESAS) to evaluate intraoperative sickness behavior. The ESAS is a common instrument to assess symptom burden in cancer patient receiving palliative care. The use of the ESAS during DBS surgery requires some rationale to follow. Please, give some reason why this instrument was chosen for this specific study.

 Answer:  We were aware that the Edmonton Symptom Assessment System (ESAS) was not a specific rating scale, which could have been applied in these conditions of awake neurosurgery. In fact, our research, as extensive as we could, did not find a more suitable scale. The ESAS, for us, could have encompassed, the different aspects of the patient's intraoperative comfort.

We add in the text why we chose this scale

 Page 3, line 127: Authors mention age and gender as secondary outcome measures. Please clarify, why age and gender were used as outcome measures? Usually, age and gender are used as covariates. Maybe, I am just wondering about the expression “outcome measure”. For me, it seems, authors just analysed whether intervention and control groups differed in terms of age, gender and baseline symptomatology. I do not consider these factors as outcome measures, but rather (sociodemographic) characteristics.

  Answer: It is correct to quote age and gender as covariates to check the initial comparability of the two arms and in and in no way as outcome measures. So we have corrected this point in the text.

Page 3, line 136: please follow one formatting style for the names of the acupuncture points (either with or without space between channel and number (e.g. “LI 4” and “P6”) see also Figure 1 as reference).

 Answer: It was done

Page 3, line 136: Please, separate “Sanyinjiao” after one Chinese word “San yin jiao” (e.g. “San- yinjiao” instead of “Sa-nyinjiao”).

 Answer:  It is true, the Chinese and international nomenclature ( WHO) separate the original name of acupuncture points in two or three parts according to the name of the points, which correspond to the meaning of the point :  SP 6  is san jin jiao. The fix have been made.

Page 3, line 137: Please, give the correct diameter of the acupuncture needles (probably “0.25mm” and not “25mm”).

   Answer: the correct diameter of the needles is 0.25 mm.

I recommend giving some rationale for the choice of the acupuncture points. Is the point selection a common set of acupuncture points for acupuncture analgesia or was it chosen for any particular reason? This could be interesting to the reader.

 Answer:  “ The choice of the acupuncture points was based on two consistent reasons:

   - all these points belong to a common set of acupuncture analgesic  points, especially in surgery, and, therefore, are frequently quoted in the publications; one of their main actions, in addition to analgesia is their regulatory action on the autonomic nervous system, especially PC 6 nei guan on nausea and vomiting;

  - the location of the chosen points correspond to their well- know cranial and CNS  therapeutic effects: headache ( LI 4  he gu, LV 3  tai chong), effect on sleep ( SP 6 san yin jiao).

 Other points could have been retained such as cranial extra meridians ( HM) points  - these latter being incompatible with this neurosurgery- or KI 6 zhao hai, but it does not seem desirable to multiply the points.”

I have included the reasons for the choice of points in the text

Results:

Page 5, line 194: authors mention the HAMA score. The HAMA (probably Hamilton Anxiety scale) is not mentioned in the methods section. Please, clarify the use of the HAMA in the methods section.

Answer:  done

    2) The secondary outcome measures were:

  • Intraoperative anesthesia monitoring parameters: heart rate, oxygen saturation, systolic and diastolic blood pressure
  • All intraoperative and postoperative adverse events were recorded.
  • Incidentally, to have a more precise approach to the patient's anxiety, we used the HAMA (Hamilton Anxiety scale) [ que je ne vois plus dans les références de cet article alors qu’elle figurait dans l’article en français «  Zigmond AS, Snaith RP. The Hospital Anxiety and Depression Scale? Acta Psychiatr Scand 1983, 67, 361-70. Doi : 10.111/j.1600-0447. 1983.tb09716.x.PMID:6880820” ]. The HAMA can be hetero- administered , as it is possible in the present study”

Page 5, line 219: I wondered whether the word “differences” is missing in this sentence? Do the authors mean to say “Significant group and time x group interaction differences were only observed for the item ‘lack of appetite’”?

Answer: The word “ differences “ has been, as it seems , omitted. I corrected it in the text.

Discussion:

Page 8, line 248: I recommend to give some interpretation of the ESAS score. What does increase of the total or a subscore mean? E.g. “…there was a significant score increase in the control versus EA group at the end of the intervention for ESAS total score and physical subscore indicating higher symptom in the control group.

Answer: the higher the ESAS score, the greater the pain or discomfortIn the control group, fatigue, pain and lack of appetite increase over time, while the acupuncturegroup sees these items decrease 

Page 8, line 249: Authors state that wellbeing increased only in the EA group. Figure 3 indicates lower wellbeing scores in the EA group compared to the control group at D2. Does a lower wellbeing score indicate higher wellbeing? This is why I recommend giving some kind of interpretation of the scores for those readers who are not familiar with the ESAS score (as I am)

 Answer : Does a lower wellbeing score indicate higher wellbeing? That’s right

Page 8, line 250: Please, double-check the sentence starting with “Finally, one patient developed…”. This sentence seems incomplete.

 Answer:  I corrected it in the text.

Page 8, line 251: Authors draw the conclusion that EA increases the probability of completing DBS procedure as one patient developed delirium during DBS surgery in the control group and the intervention was stopped. I do not follow this statement as the number of n=1 is too small to draw any firm conclusions. Please, reword this sentence to a more cautious conclusion.

Answer: the reviewer is right. I modified the sentence.

The authors should list some more limitations of the study at hand such as small sample size, single study center, lack of blinding, lack of a sham intervention, lack of a firm primary hypothesis being tested…

 Answer :

   - small sample size, mainly due to the calculated number of patients to get a significative result according to the ESAS score,

    - single study center due to the high specificity of this surgery and the availability of acupuncturists

to  to work

    - the feasibility of blinding and sham intervention in acupuncture randomized controlled clinical studies have been widely discussed , considering technical and physiological aspects of acupuncture

All these sentences were added in the text.

Reviewer 2 Report

I have two observations regarding this study that should be addressed by the authors:

Firstly, statistical results reported are very vague, as they include p-values only, totally missing the presentations of the statistics (such as F), Degrees of Freedom, confidence intervals, and, crucially, effect sizes. Being the later the most important measure in this type of studies, as relying in p-values only could potentially lead to misleading conclusions. In addition, there is no mention of data screening whatsoever.

Secondly, my main concern is the actual design of this experiment. It is obvious that making the intervention subjects “blind” from the treatment is not feasible, and therefore (considering also the subjectivity of the measurement instrument) the results may not be statistically sound. The point is, considering one intervention only, the subjects under study may feel better just because they “know” that were treated with EA, not because of the EA intervention per se. Maybe a second type of intervention would be necessary.

An extra point to add here is the one arguing that EA “increases the probability of completing the DBS procedure” (line 251). This conclusion is misleading as the only one patient developing delirium may have happened just by chance.

Author Response

Open Review

English language and style

( ) Extensive editing of English language and style required
( ) Moderate English changes required
(x) English language and style are fine/minor spell check required
( ) I don't feel qualified to judge about the English language and style

Yes

Can be improved

Must be improved

Not applicable

Does the introduction provide sufficient background and include all relevant references?

(x)

( )

( )

( )

Is the research design appropriate?

( )

( )

(x)

( )

Are the methods adequately described?

( )

(x)

( )

( )

Are the results clearly presented?

( )

( )

(x)

( )

Are the conclusions supported by the results?

( )

( )

(x)

( )

Comments and Suggestions for Authors

I have two observations regarding this study that should be addressed by the authors:

Firstly, statistical results reported are very vague, as they include p-values only, totally missing the presentations of the statistics (such as F), Degrees of Freedom, confidence intervals, and, crucially, effect sizes. Being the later the most important measure in this type of studies, as relying in p-values only could potentially lead to misleading conclusions. In addition, there is no mention of data screening whatsoever.

Answer: The missing data were handled by using the Last-Observation-Carried-Forward method. The significance threshold was set at 5%. In case of significance of time interaction, Dunnett's multiple comparisons post-hoc test was used, comparing data recorded at all the timepoints in each group (control or EA) with respect to the initial timepoint (incl) taken as a reference. In case of significance of group interaction, Sidak's multiple comparisons post-hoc test was used to compare the data recorded in the two groups (EA and control) at all the timepoints.

As secondary outcome measures, intraoperative anesthesia monitoring parameters (heart rate, oxygen saturation, and systolic and diastolic blood pressure) were analyzed with the same repeated-measures ANOVA, but only considering intraoperative time points (from inst to end).

A comparison between the two groups (control and EA) at baseline was performed using the Mann-Whitney test for age, UPDRS score, total ESAS score and physical and emotional ESAS subscores and the Fisher’s exact test for gender. A comparison between the two groups regarding the total time duration of the surgery was also performed using the Mann-Whitney test. Prism statistical software (GraphPad Software, San Diego, CA) was used in all cases.

Statistical analysis were performed by Pr Jean Michel Nguyen as defined initially by the protocol. Maybe we can do more statistical analysis but we choose not to modify the protocol.

In this study, the global score of the “Edmonton Symptom Assessment Scale” is retained since it totals all the items testifying to the patient’s comfort. This scale has an amplitude of 90 points spread over 9 scales. In a published study [10], the standard deviation was estimated at 9.5. We propose to retain a larger standard deviation (SD = 11) because our population is a little different from this study. We predict that the control group will have a score of 50 on a 90-point scale and that the intervention would reduce this score by 38%, or 31 points, which corresponds to our experience.

Taking an alpha risk of 5% and a power of 90%, we would need 8 patients in each arm to demonstrate this decrease in the overall score.

In order to compensate for a possible decrease in the power of the non-parametric test (Wilcoxon) compared to the Student test, 2 additional patients will be included in each group, i.e. 10 per group. The theoretical power would then be 96% according to the Student test.

This recruitment would also correspond to one year of recruitment for the service.

Secondly, my main concern is the actual design of this experiment. It is obvious that making the intervention subjects “blind” from the treatment is not feasible, and therefore (considering also the subjectivity of the measurement instrument) the results may not be statistically sound. The point is, considering one intervention only, the subjects under study may feel better just because they “know” that were treated with EA, not because of the EA intervention per se. Maybe a second type of intervention would be necessary.

    Answer :

 - the patients were randomly assigned to this study,

       - about the sample size , this small sample size is  mainly due to the calculated number of patients to get a significative result according to the ESAS score,

      - the feasibility of blinding and sham intervention in acupuncture randomized controlled clinical  

studies have been widely discussed , considering technical and physiological aspects of acupuncture.

  Taking in account the present data on this subject, in the literature, we made the evaluation to be performed not by any therapist (neurosurgeon, acupuncturist, anesthesiologist) but carried by a third party, who was a medical assistant

   - about the placebo effect of acupuncture, we are aware of it, as there is a placebo effect in any treatment, especially physical treatment: however our results showing sometimes better ESAS score in the non - acupuncture arm, do not support a single placebo effect.

  - anyway, a second type of intervention, with three arms, is foreseen

Before this study we performed a non-randomized study on 10 PD patients who will not underwent surgery without acupuncture because they were too anxious of the surgery. All patients can have the surgery and I give you the results of the poster presented to 16th congress of Iasp, 26-30 september Yokohama 2016. Results were good with decreased of pain on Vas than more 50% compared to control subjects, anxiety was decreased too and patient’s satisfaction was evaluated with 80% of patients said that pain VAS was about 2/10 and decrease of anxiety score.All these patients could have a strong placebo effect, which can modify the interpretation of results; In the study we presented here, we can have placebo or nocebo effect because the subjects are randomized.I agree with the reviewer that there is subjectivity of the measurement instrument, but it is the case in all pain studies and we must deal with that even if scales, evaluations are not perfect. It is very difficult to have strong statically studies on pain because of the  subjectivity of measurement but we tried with this randomized, control, prospective study to avoid the maximum of bias we can.

Round 2

Reviewer 2 Report

Despite my comments about weaknesses while presenting the statistical results, it seems that the authors were not able to properly address them. Furthermore, regarding my questioning on the feasibility of alternatives for blinding (such as a different type of treatment that could lead to the same results, not because the treatment per se but because the existence of an intervention only) is addressed by the authors as “widely discussed” without given proper references to sustain their claim.

Author Response

Open Review

English language and style

( ) Extensive editing of English language and style required
( ) Moderate English changes required
(x) English language and style are fine/minor spell check required
( ) I don't feel qualified to judge about the English language and style

Yes

Can be improved

Must be improved

Not applicable

Does the introduction provide sufficient background and include all relevant references?

(x)

( )

( )

( )

Is the research design appropriate?

( )

( )

(x)

( )

Are the methods adequately described?

( )

(x)

( )

( )

Are the results clearly presented?

( )

( )

(x)

( )

Are the conclusions supported by the results?

( )

( )

(x)

( )

Comments and Suggestions for Authors

I have two observations regarding this study that should be addressed by the authors:

Firstly, statistical results reported are very vague, as they include p-values only, totally missing the presentations of the statistics (such as F), Degrees of Freedom, confidence intervals, and, crucially, effect sizes. Being the later the most important measure in this type of studies, as relying in p-values only could potentially lead to misleading conclusions. In addition, there is no mention of data screening whatsoever.

The presentation of the statistics has been completed: F values, df, CI and effect sizes have been added, including the interpretation of the effect sizes in the Results (page 6, lines 217 – 247).

Secondly, my main concern is the actual design of this experiment. It is obvious that making the intervention subjects “blind” from the treatment is not feasible, and therefore (considering also the subjectivity of the measurement instrument) the results may not be statistically sound. The point is, considering one intervention only, the subjects under study may feel better just because they “know” that were treated with EA, not because of the EA intervention per se. Maybe a second type of intervention would be necessary.

The issue of blinding (and sham procedure) has been addressed in the Discussion (page 9, lines 314 – 327).

An extra point to add here is the one arguing that EA “increases the probability of completing the DBS procedure” (line 251). This conclusion is misleading as the only one patient developing delirium may have happened just by chance.

Reviewer 2 is right, and we have already answered to Reviewer 1 on this point: this sentence has been reworded more cautiously (page 8, lines 287 - 288).
